# Empirical Gateaux Derivatives for Causal Inference

**Michael I. Jordan**[*]
Department of EECS and Department of Statistics
University of California, Berkeley
jordan@cs.berkeley.edu

**Yixin Wang**[*]
Department of Statistics
University of Michigan
yixinw@umich.edu

**Angela Zhou**[*]
Department of Data Sciences and Operations
University of Southern California
zhoua@usc.edu

## Abstract

We study a constructive algorithm that approximates Gateaux derivatives for statistical functionals by finite differencing, with a focus on functionals that arise in causal inference. We study the setting where probability distributions are not known *a priori* but need to be estimated from data. These estimated distributions lead to empirical Gateaux derivatives, and we study the relationships between empirical, numerical, and analytical Gateaux derivatives. Starting with a case study of the interventional mean (average potential outcome), we delineate the relationship between the empirical Gateaux derivative (via finite differencing) and the analytical Gateaux derivative. We then derive requirements on the rates of numerical approximation in perturbation and smoothing that preserve the statistical benefits of one-step adjustments, such as rate double robustness. We further study more complicated functionals such as dynamic treatment regimes and the linear-programming formulation for policy optimization in infinite-horizon Markov decision processes. The ability to approximate bias adjustments in the presence of arbitrary constraints in these more complicated settings illustrates the usefulness of constructive approaches for Gateaux derivatives. We also find that the statistical structure of the functional (rate double robustness) can permit less conservative rates for finite-difference approximation. This property, however, can be specific to particular functionals; e.g., it occurs for the average potential outcome (hence average treatment effect) but not the infinite-horizon MDP policy value.

## 1 Introduction

Inferential targets in causal machine learning often take the form of statistical functionals of the data distribution. Examples include average treatment effects or average policy values for infinite-horizon offline reinforcement learning. Estimation of these statistical functionals can be vulnerable to the first-stage bias introduced by the estimation of nuisance functions such as outcome regressions or transition probabilities. However, by leveraging the causal structure, it is possible to derive bias-adjusted estimators that weaken the need for accurate estimation of these nuisance functions. The celebrated doubly robust estimator is one such bias adjustment, and there are general frameworks for deriving such estimators, including semiparametric efficiency, double robustness, Neyman orthogonality, and "debiased/double" machine learning [7, 12, 13, 35, 52, 53, 56, 62]. These general frameworks can also be applied to more complicated functionals in longitudinal causal

---

[*]Authors listed in alphabetical order.

inference and offline reinforcement learning (also known as offline policy evaluation/learning) [5, 11, 43, 46–49, 67, 68, 71].

A drawback of these general frameworks is that significant analytic effort is often necessary to obtain concrete estimators that are appropriate for particular situations. Indeed, a practitioner may be interested in novel variants of an estimand, or be working within a constrained class of functionals that are appropriate for a particular class of problems. For example, in *constrained* Markov Decision Processes (MDPs) [3], or optimization-based estimators more broadly, a practitioner can easily avail themselves of custom convex-optimization solvers that are computationally efficient and easy to deploy to add additional constraints. Unfortunately, such choices may require case-specific re-analysis to establish the desired statistical properties, and deriving the actual estimator that yields bias adjustment may require significant analytical effort. It is therefore important to develop a suite of constructive, numerical or algorithmic methods that yield desired forms of bias adjustment. Such a suite would be complementary to analytic derivations.

To this end, we develop off-the-shelf procedures for estimating the statistical functionals for bias adjustment in causal inference. We do so via numerical approximation of the Gateaux derivatives that underlie the general analytic frameworks, building on prior work in this area. We focus on interventional effects. [57] suggested numerical differentiation to estimate Gateaux derivatives of functionals unavailable in closed form but approximated by computational procedures, such as the solution to a system of differential equations; this approach is later used in [34]. Recent pedagogical surveys of [35, 53] also emphasize the usefulness of Gateaux derivatives for deriving influence functions, an approach which is directly applicable for discrete data and generalizes to continuous distributions via a smoothing argument as discussed in [10, 39]. In this spirit, we build on the line of work of Carone et al. [10], Frangakis et al. [24], Ichimura and Newey [39], Newey [57], which proposes a constructive procedure by numerical approximation of a Gateaux derivative (in the sense of Hampel [32], Huber [38]) by finite differences.

Evaluating influence functions by numerical differentiation can be useful because it only requires black-box evaluation of the functional. In other areas such as optimization, numerical differentiation underpins common subroutines that enable analysts to use optimization algorithms without specialized training in a particular modeling language or paradigm.[2] Although more advanced paradigms compute exact gradients, they can require specialized training to use. In addition, the numerical differentiation viewpoint connects the conceptual interpretation of influence functions as qualitative sensitivity analysis (i.e. sensitivity of a functional to perturbations in the distribution) to the use of influence functions in causal inference. Typical presentations of the latter framework are not concretely connected to the former notion. We provide a concrete connection, which can be interesting independent of computerized estimation for algorithmic, pedagogical, or conceptual reasons.

In this paper, we build on the previously mentioned line of work on the use of numerical derivatives to approximate influence functions, focusing on the implications for statistical estimation. We provide exact characterizations of empirical Gateaux derivatives, in contrast to numerical derivatives computed based on oracle knowledge of probability distributions. This work is focused on exact characterizations that can inform statistical and computational trade-offs. Although we expect that some of these characterizations may be apparent to experts, we also hope that this level of concreteness can be helpful to non-experts. These trade-offs could eventually be practically relevant because the finer-scaled discretization required for lower approximation error can incur fundamental issues with floating-point computation that prohibit further gains in accuracy [23, 64].

Our contributions are as follows. We begin with a case study of the average potential outcome. The average treatment effect is the difference of the average potential outcomes, and our analysis applies also to the average treatment effect. We discuss the treated mean for simplicity; by symmetry the same results hold for the control mean. In this setting, we characterize the exact relationship between finite-differences and the analytical Gateaux derivative. This characterization helps concretize the proposal of empirical Gateaux derivatives by allowing us to study the rates of *numerical* approximation that can preserve *statistical* performance. Finally, with these concretizations in hand, we turn to more complicated motivating examples including a finite-horizon dynamic treatment regime and infinite-horizon reinforcement learning. We illustrate with the case of policy learning in finite-state,

---

[2]For example, many data scientists use optimization through the `scipy.optimize` or `r.optim` packages: if they do not provide a gradient function callback, in some configurations, it is by default approximated by numerical differentiation.

finite-action infinite-horizon Markov decision processes how this constructive approach to Gateaux derivatives can directly extend also to the case of custom constraints.

## 2  Problem Setup: From Numerical to Empirical Gateaux Derivatives

We begin by introducing our first example, that of the mean potential outcome, and the canonical doubly-robust estimator. We then briefly introduce the key objects in the more general framework of orthogonalizing or debiasing statistical functionals by Gateaux derivative adjustments via the one-step estimator. We also describe the numerical approximation of these adjustments via perturbed black-box evaluations of the statistical functional and plug-in estimation. After introducing these frameworks, we clarify what we mean by *empirical* rather than *numerical* Gateaux derivatives and delineate our specific research questions.

We let $O \sim P$ denote a draw of observations following the distribution $P$ belonging to a statistical model $\mathcal{M}$. We will focus on the estimation of statistical functionals $\Psi(P)$, where $\Psi : \mathcal{M} \to \mathbb{R}^q$ is pathwise differentiable. Throughout, we take as given the assumptions that grant causal identifiability (i.e., overlap/positivity, unconfoundedness/ignorability).

**The mean potential outcome and augmented inverse propensity weighting.**  We very briefly overview the celebrated doubly robust estimate of the mean potential outcome. The average treatment effect is simply the difference of these doubly robust estimators for the treated and control means; for simplicity, we discuss the treated mean only.

**Example 1** (Mean potential outcome)**.** *The observation $O = (X, A, Y)$ includes covariate $X$, treatment $A \in \{0, 1\}$, and outcome $Y \in \mathbb{R}$. The statistical functional corresponding to the mean potential outcome is:*

$$\Psi(P) = \mathbb{E}[Y(1)] = \mathbb{E}[\mathbb{E}[Y \mid A = 1, X]].$$

Plug-in estimation of the conditional expectation is termed *regression adjustment*. The overall functional is also referred to as *inverse propensity weighting,* $\Psi(P) = \mathbb{E}[Y(1)] = \mathbb{E}\left[Y\frac{\mathbb{I}[A=1]}{e(X)}\right]$, where $e(x) = P(A = 1 \mid X = x)$ is the *propensity score*. (We omit explicit discussion of assumptions such as consistency and ignorability, under which these identification results hold).

Given two identifying functionals for the mean potential outcome, a natural question is: can these identification approaches be combined in some way to improve estimation? The *doubly-robust* estimator of Robins et al. [62] achieves this as follows:

$$\mathbb{E}[Y(1)] = \mathbb{E}\left[\frac{\mathbb{I}[A=1]}{p(A=1|X)}(Y - \mathbb{E}[Y \mid A = 1, X]) + \mathbb{E}[Y \mid A = 1, X]\right]$$

Appealing properties such as the mixed-bias property and the rate double robustness properoty can be verified readily from its functional form. This provides a canonical example of one-step adjustment by influence functions that applies more broadly.

**Influence function and Gateaux derivative.**  A functional $\Psi$ is *Gateaux differentiable at $P$* if $\Psi'(H; P) \triangleq \frac{d\Psi(P+\epsilon H)}{d\epsilon}\big|_{\epsilon=0}$ exists and is linear and continuous in $H$ for all $H$.[3]

Hampel [32] and Huber [38] define the *influence function $\phi(O; P)$* as a Gateaux derivative with respect to perturbations $H = \delta_o - P$, where $\delta_o$ is a degenerate Dirac measure satisfying the identity $\int o\delta_{o'}(o) = o'$. That is, we have:

$$\phi(o; P) = \frac{\mathrm{d}\Psi\left(P + \epsilon(\delta_o - P)\right)}{\mathrm{d}\epsilon}\bigg|_{\epsilon=0}.$$

**Numerical Gateaux derivatives.**  Although this definition of influence function via perturbation is not quite enough for semiparametric inference with continuous distributions, Carone et al. [10] and Ichimura and Newey [39] show how to obtain absolute continuity in semiparametric models by taking an additional limit as a *smoothed perturbation* converges to the point mass. They replace the Dirac delta measure $\delta_o$ with a smoothed distribution $\tilde{\delta}_{o'}^{\lambda}(o)$, a function satisfying $\int g(u)\mathrm{d}\tilde{\delta}_{o'}^{\lambda}(u) = 1$

---

[3]Another way to state the definition is that $\Psi$ is Gateaux differentiable at $P$ if there exists $\phi_P(H)$ such that $\phi(\theta + \epsilon H) - \Psi(P) = \epsilon\phi_P(H) + o(t)$, as $t \to 0$ [73, p.296]. This statement coincides with the previous statement when the functional maps to the real line.

and dominated by $P$. A common choice for $\tilde{\delta}_{o'}^{\lambda}$ has the form $\tilde{\delta}_{o'}^{\lambda}(o) = K_{\lambda}(o - o')$, where $K_{\lambda}$ is a bandwidth-normalized kernel function and product kernel. We then define a perturbation in the direction of a generic observation $o_i$ as follows: $P_{\epsilon,\lambda}^{o_i} = P_{\epsilon,\lambda}^i = (1 - \epsilon)P + \epsilon\tilde{\delta}_{o_i}^{\lambda}$.

Carone et al. [10], Frangakis et al. [24] discuss *numerical* approximation by finite differences: for a fixed $\epsilon, \lambda$ with $\epsilon \ll \lambda$, evaluate the numerical derivative $\hat{\phi}(o; P) = \epsilon^{-1}(\Psi(P_{\epsilon,\lambda}) - \Psi(P))$. Further, [35, 53] highlight influence function derivations based on the analytic Gateaux derivative.

Carone et al. [10] establish that the limit of $P_{\epsilon,\lambda}^{o_i}$ when $\epsilon$ and $\lambda$ tend to zero is precisely the Gateaux derivative. and they provided generic approximation rates in $(\epsilon, \lambda)$ for a multi-point finite difference scheme. These rates establish sufficient conditions for approximation error that ensure validity of the numerical approximation, but can be conservative. For a two-point approximation with a uniform kernel, the rate from Carone et al. [10, Thm. 5] is $(\epsilon\lambda^{-d}) + \lambda^{-2}$. We later strengthen this result.

**Empirical Gateaux derivatives.** In the context of statistical estimation, the underlying probability distributions also need to be estimated. We consider *plug-in estimation* of the statistical functional via density estimation of the collection of joint probability estimates $\tilde{p}(o)$ that comprise $\tilde{P}$. For example, a plug-in representation of the mean potential outcome is as follows: $\Psi(\tilde{P}) = \int \int y \frac{\tilde{p}(y, A=1, x)}{\tilde{p}(A=1, x)} \tilde{p}(x) \mathrm{d}y\mathrm{d}x$.

It is important to distinguish between the *empirical Gateaux derivative* computed by numerical approximation with estimated probability densities (e.g., $\tilde{P}(Y, A = 1, X)$), the *numerical Gateaux derivative* obtained by finite differencing based on the true probability densities, and the analytic Gateaux derivative:

$$\tilde{\phi}(O_i) = \epsilon^{-1}\left(\Psi(\tilde{P}_\epsilon^i) - \Psi(\tilde{P})\right) \quad \text{empirical derivative at smoothed and estimated distributions,}$$

$$\hat{\phi}(O_i) = \epsilon^{-1}\left(\Psi(P_\epsilon^i) - \Psi(P)\right) \quad \text{numerical derivative at smoothed and true distributions,}$$

$$\phi(O_i) = \left.\frac{\mathrm{d}}{\mathrm{d}\epsilon}\Psi(P_\epsilon^i)\right|_{\epsilon=0} \quad \text{analytical Gateaux derivative.}$$

It is the first of these representations that is appropriate for serving as a statistical estimator of influence functions. In particular, the *one-step estimator* is justified by the following asymptotically linear representation [60]:

$$\Psi_n = \Psi(\tilde{P}) + \tfrac{1}{n}\sum_{i=1}^n \tilde{\phi}(O_i), \qquad \text{where } \tilde{\phi}_i(O_i) = \tfrac{1}{\epsilon}(\Psi(\tilde{P}_\epsilon^i) - \Psi(\tilde{P})). \tag{1}$$

See Appendix A for further discussion of this expansion. Given these definitions, we now turn to the major mathematical and statistical questions that we address in this paper:

**Q1**: How exactly does the empirical Gateaux derivative approximate the analytical derivative?

**Q2**: What are the required rates of *numerical approximation* in perturbation and smoothing based on $(\epsilon, \lambda)$ that preserve the beneficial *statistical* properties of the constructed estimator?

In Section 4, we study these questions in the simple case of our running example: estimation of the mean potential outcome. In that setting we can provide a fine-grain comparison of the finite-difference estimator and the classical doubly robust estimator. In Section 5, we study more complicated functionals to highlight more general applicability.

**Example 2** (Example 1, continued). *The influence function [31, 62] for a mean potential outcome is:*

$$\phi(O) = \frac{\mathbb{I}[A=1]}{p(A=1|X)}(Y - \mathbb{E}[Y \mid A = 1, X]) + \mathbb{E}[Y \mid A = 1, X] - \Psi(P).$$

Hence the canonical doubly-robust estimator (augmented inverse propensity weighting, or AIPW) can be viewed as a one-step adjustment.

We conclude our introductory remarks by noting some limitations of our study. We omit discussion of semiparametric efficiency since we focus on computing influence functions without the consideration of model restrictions and tangent spaces. That is, we focus on Gateaux derivatives under a nonparametric model. Throughout, we assume the functional is pathwise differentiable. This assumption does warrant caution, because this condition can pose a fundamental barrier to efforts for

"automation"; hence empirical Gateaux derivatives are intended to augment but not supplant or replace analytical expertise. See Appendix D.2 for extended discussion on how stronger smoothness conditions admit sufficient conditions for asymptotic linearity.

# 3  Related Work

In this section, we discuss the most closely related directions but do not attempt to provide a survey or overview of influence functions or semiparametric statistics. See Bickel et al. [7], Fisher and Kennedy [21], Kennedy [53], Tsiatis [69] or the appendix for a broader overview and further details. In Appendix D.1, for additional context, we also overview work on numerical derivatives in optimization and statistical machine learning more broadly.

**Numerical approximation of Gateaux derivatives.** Recent pedagogical reviews [21, 35, 53] emphasize the Gateaux derivative and its use as a heuristic tool for deriving influence functions. While our point of departure is a finite-difference approach to the former, we also consider more complicated optimization functionals, with a focus on how dual variables characterize the Gateaux derivatives therein. [10] derive high-level sufficient conditions for approximation error via general results for finite-difference approximations. Relative to those general sufficient conditions, we focus on specific improvements in approximation error for various functionals. For the specific case of the mean potential outcome/average treatment effect, Frangakis et al. [24] interchange integration and differentiation as a result of analytical insight hence their approach is less "automatic." In general, it may be difficult to deduce the form of the nuisance functions without deriving the influence function analytically. The approach in Appendix B of [9] considers the specific case of debiasing moment conditions with respect to first stages but also implicitly leverages this change in order of integration.

**Other work on automating causal inference.** Recent work in causal inference aims at "automatic" or "computerized" semiparametric inference via a variety of methods [4, 14–16, 20, 45, 51]. To summarize briefly the difference with our approach, our approach focuses on the use of numerical derivatives, and requires *only* a zeroth-order computation oracle of perturbed functional evaluations, without additional nuisance estimation. This approach can be particularly helpful for incorporating debiased causal inference in the design and analysis of algorithms. On the other hand, it may not achieve as strong improvements in estimation performance as other approaches that estimate additional nuisance functions. Concurrently with the initial submission of this work, [66] studied automatic debiased machine learning for dynamic treatment effects. Our development is most distinctive in the infinite-horizon setting, which is not a generalized regression residual.

We discuss differences to other works in the line of work of "automatic" semiparametric estimation that are less directly related in Appendix A.1.

# 4  Empirical Gateaux Derivative of the Mean Potential Outcome

To illustrate the numerical approach to Gateaux derivatives, we work through the example of the mean potential outcome and indicate precisely how the *numerical differentiation* incurs error relative to the analytical derivative, which here is the doubly robust or augmented IPW (AIPW) estimator.

**Relating the empirical and analytical Gateaux derivatives**

**Setup.** We target the estimation of a mean potential outcome as in Example 1. For the smoothed perturbation, we specialize to kernel functions $K_\lambda(u)$ from kernel density estimation satisfying $\int_{-\infty}^{\infty} K(u)du = 1$ (normalization to a probability density) and $K(-u) = K(u)$, for all $u$ (symmetry).[4] The mean potential outcome is identified as follows,

$$\Psi(P) = \mathbb{E}[Y(1)] = \mathbb{E}[\mathbb{E}[Y \mid A = 1, X]] = \int \int y p(y \mid A = 1, x)\mathrm{d}y\mathrm{d}x = \int \int y \frac{p(y, A=1, x)}{p(A=1, x)} p(x)\mathrm{d}y\mathrm{d}x.$$
(2)

We obtain $\Psi(\tilde{P}_\epsilon)$ by plugging in the $\tilde{p}$ probability density estimates. Note this is distinct from the plug-in estimator in empirical process theory, which plugs in the empirical CDF; we plug in statistical estimates. See Algorithm 1 for a summary of the procedure.

---

[4]Examples include the Gaussian kernel with $K(u) = (2\pi)^{-\frac{1}{2}} e^{-u^2/2}$, or uniform with $K(u) = 1/2\mathbb{I}\left[|u| \leq 1\right]$. We generally consider $K_\lambda(u) = \lambda^{-d} K(u/\lambda), K(x) = K(x_1, \ldots, x_d)$.

**Comparison of empirical Gateaux derivative and augmented inverse propensity weighting.**
We have argued that, in the context of statistical estimation, approximating Gateaux derivatives requires plug-in estimation of the statistical functional with estimated probabilities. Towards resolving Q1, we observe that the perturbed probability densities induce the functional form of the nuisance functions in the analytical derivative. This is in contrast to the analytical derivative, which describes the estimand in terms of, e.g., conditional expectations, and allows the analyst to choose the functional form for estimating the nuisance function.

We will denote the $\tilde{P}$-induced conditional expectation as

$$\mathbb{E}_{\tilde{P}}[Y \mid A = 1, X = x] = \int y \tilde{p}(y \mid A = 1, x)\mathrm{d}y.$$

For example, the density-induced conditional expectation of kernel density estimates is exactly Nadaraya-Watson regression. Given this, we study empirical derivatives with kernel density estimation because they provide a classic textbook example of density estimates that induce conditional expectation estimators. Other density estimation approaches can be used and their rates of convergence would simply be substituted in Lemma 1. In Appendix D we discuss other choices of density estimates. Of course, all such nonparametric estimation approaches suffer from the curse of dimensionality, so additional structure is required to satisfy the product-rate conditions.

Our characterization will show exactly how plug-in evaluation with finite differences is nearly equivalent to evaluating the doubly-robust estimator with a Nadaraya-Watson regressor as the conditional outcome regression for $\mathbb{E}[Y \mid A = 1, X]$ and a kernel density estimate of the propensity score. It is equivalent up to an additive bias that arises due to the smoothed perturbation distribution, and what we will term a *smoothed nuisance evaluation*. We discuss what this smoothed nuisance function is, and how we study its approximation error relative to estimation error of the induced nuisance.

**Integrating the smoothed perturbation induces smoothed nuisance functions.**   Integrating with respect to the smoothed perturbation distribution can be interpreted as a smoothed evaluation of the resulting nuisance functions. We define the $\tilde{\delta}_{x_0}^{\lambda}$-smoothed conditional expectation which smooths the evaluation point, i.e., smooths evaluation around $x_0$ rather than precisely at $x_0$.

**Definition 1** ($\tilde{\delta}_{x_0}^{\lambda}$-smoothed conditional expectation)**.**

$$\tilde{\mathbb{E}}_P[Y \mid A = 1, X = x_0] = \int \mathbb{E}_P[Y \mid A = 1, X = u]\, \tilde{\delta}_{x_0}^{\lambda}(u)\, \mathrm{d}u. \tag{3}$$

A similar definition appears in the literature on nonregular inference or the localization of a global functional [57].[5] The following lemma, similar to van der Laan et al. [72, Lemma 25.1], summarizes convergence rate implications of these smoothed nuisances; these are direct consequences of typical analyses of kernel density estimation.

**Lemma 1.** *Let $\tilde{\mu}_\epsilon(X) = \tilde{\mathbb{E}}_{\tilde{P}_\epsilon}[Y \mid A = 1, X]$ and $\tilde{e}_\epsilon(X) = \tilde{p}_\epsilon(A = 1, X)/\tilde{p}(X)$ denote the nuisances induced by plug-in estimation of probability distributions $\tilde{p}$. For any bounded function $g(x)$ in a Hölder class of degree $\beta$, $(\int \tilde{\delta}_x^{\lambda}(u)g(u)\,\mathrm{d}u - g(x))^2\,\mathrm{d}x = O(\lambda^\beta)$. Assume that $\tilde{\delta}^\lambda$ is a product kernel in a Hölder class of degree $\beta$, and $\int |u|^\beta |\tilde{\delta}_u^\lambda|\,\mathrm{d}u < \infty$ and $\int u^s \tilde{\delta}_u^\lambda\,\mathrm{d}u = 0$ for $s \leq \beta$. Assume $Y$ is bounded. Then the perturbed nuisances satisfy the following rates:*

$$\|\tilde{\mu}_\epsilon(X) - \mu(X)\| = \|\tilde{\mu}(X) - \mu(X)\| + O(\epsilon\lambda^{-d/2}),$$
$$\|\tilde{e}_\epsilon(X) - e(X)\| = \|\tilde{e}(X) - e(X)\| + O(\epsilon\lambda^{-d/2}).$$

The perturbed nuisances are asymptotically consistent, so long as $\tilde{p}$ induces asymptotically consistent nuisances and we fix $\epsilon$ as a sequence vanishing in $n$ at a rate appropriate to counteract the growth in $\lambda$ due to "roughness"; i.e., the variability of $\tilde{\delta}^\lambda$. The difference between the function evaluation and its smoothed nuisance arises from the bias analysis of kernel density estimators and the dimension-dependence of ensuring absolute continuity, but it is a non-stochastic, entirely deterministic argument. While the analysis improves mildly in the dimension dependence on $\lambda$ compared to the general analysis of Carone et al. [10, Thm. 5], this still highlights the generally unfavorable dependence of the numerical approach on the dimension due to smoothing.

---

[5]For example, van der Laan et al. [72] proposes smoothing a nonregular target estimand, such as a dose-response curve or density evaluated at a point $x_0$, by conducting a smoothed evaluation locally around $x_0$. In that work, a function evaluated at $x_0$, $g(x_0)$, is approximated with a kernel smooth over $x$ in a neighborhood of $x_0$ with bandwidth $h$; so $\Psi_\lambda(x_0) = \int K_\lambda(u - x_0)g(u)\mathrm{d}u$. Jung et al. [44] use a similar smoothing/localization device but for a different problem.

---
**Algorithm 1** Empirical Gateaux derivatives
---
1: Inputs: $\tilde{P}$ probability density estimates, data $\{O_i\}_{1:n}$, perturbation and smoothing $(\epsilon, \lambda)$ parameters, functional $\Psi$
2: **for** $i = 1, \ldots n$ **do**
3: $\quad \tilde{P}_\epsilon^i = (1 - \epsilon)\tilde{P} + \epsilon \tilde{\delta}_{o_i}^\lambda$
4: $\quad \tilde{\phi}(O_i) \leftarrow \epsilon^{-1}(\Psi(\tilde{P}_\epsilon^i) - \Psi(\tilde{P}))$
5: **end for**
6: Output: one-step adjusted estimate from empirical Gateaux derivatives, $\Psi(\tilde{P}) + \frac{1}{n}\sum_i \tilde{\phi}(o_i)$
---

Our first proposition establishes the exact approximation error induced by finite differencing.

**Proposition 1** (Gateaux derivative of probability density representation)**.** *Consider perturbations in the direction of $o_i = (x_i, a_i, y_i)$. Let $\tilde{p}(x), \tilde{p}(A = 1, x), \tilde{p}(y, A = 1, x)$ denote kernel density estimates. Let $\Psi(P)$ be as in Equation* (2)*:*

$$
\frac{\Psi(\tilde{P}_{\epsilon_i}) - \Psi(\tilde{P})}{\epsilon} = \int \frac{\tilde{p}(x)\mathbb{I}[a_i = 1]\tilde{\delta}_{x_i}^\lambda(x)}{\tilde{p}_\epsilon(A = 1, x)} \left\{ \left( \int y\, \tilde{\delta}_{y_i}^\lambda(y)\mathrm{d}y \right) - \mathbb{E}_{\tilde{P}}[Y \mid A = 1, X = x] \right\}\, \mathrm{d}x
$$
$$
+ \left( \tilde{\mathbb{E}}_{\tilde{P}_\epsilon}[Y \mid A = 1, X = x_i] - \Psi(\tilde{P}) \right).
$$

We compare the expansion of Proposition 1 to the canonical AIPW estimator in Example 2. We specialize to a uniform kernel for $\tilde{\delta}_{y_i}^\lambda(y)$ so that $\int y\, \tilde{\delta}_{y_i}^\lambda(y)\mathrm{d}y = y_i$. We next observe that the smoothing kernel introduces additive bias relative to evaluating terms in the expression at the observation $o_i$.

**Corollary 1.** *When the perturbation observation is the observation datapoint $O_i = (X_i, A_i, Y_i)$,*

$$
\tilde{\phi}(O_i) = \frac{\mathbb{I}[A_i = 1]}{\tilde{p}_\epsilon(A = 1 \mid X_i)}\left(Y_i - \mathbb{E}_{\tilde{P}}[Y \mid A = 1, X_i]\right) + \left( \mathbb{E}_{\tilde{P}_\epsilon}[Y \mid A = 1, X_i] - \Psi(\tilde{P}) \right) + O(\lambda^\beta).
$$

Lemma 1 allows us to further simplify and conclude that the decomposition of Proposition 1 is close to the canonical AIPW estimator up to the $O(\lambda^\beta)$ bias induced by smoothed evaluation.

**Implications for estimation**  Next, we use our characterization in Proposition 1, Lemma 1, and Corollary 1 to study the *statistical* properties of our *computational/numerical* approximation. Corollary 1 arises from deterministic equivalences and allows us to study the relationship to the typical "oracle" AIPW estimator; albeit with induced $\epsilon$-perturbed nuisances. An appealing property of bias-adjusted treatment effect estimation is *rate double robustness*: because we incur the product of convergence rates of the nuisances, we may enjoy parametric $n^{-\frac{1}{2}}$ rate convergence of the target functional while nuisances converge at slower rates, for example at $n^{-\frac{1}{4}}$; see [13, 63]. We use the rate double robustness property of the target and Lemma 1 to infer the required rate conditions on $(\epsilon, \lambda)$ that retain the beneficial statistical properties of the canonical AIPW estimator. The next result combines our previous characterizations with the standard analysis of AIPW.

**Assumption 1** (Regularity conditions)**.** *Assume the following regularity conditions hold:*

(i) *$Y$ is bounded.*

(ii) *$p(y \mid A = 1, x), p(A = 1 \mid x)$ are in Hölder classes of minimum degree $\beta$.*

(iii) *The estimates $p(y, a, x), p(a, x), p(x)$ belong to a Donsker function class.*

(iv) *Assume $\tilde{\mu}, \tilde{e}$ satisfy the product-rate condition: $\|\tilde{\mu}(X) - \mu(X)\| \times \|\tilde{e}(X) - e(X)\| = O_p(n^{-\frac{1}{2}})$. Assume $\tilde{\mu}, \tilde{e}$ are RMSE-consistent with $r_\mu, r_e$, respectively, so that $r_\mu + r_e \geq \frac{1}{2}$.*

**Theorem 1** (Rate double robustness)**.** *Consider the one step-estimator with the empirical Gateaux derivative. Under Assumption 1, when $\epsilon\lambda^{-d/2} = o(n^{-\max(r_\mu, r_e)})$, and $\lambda^\beta = o(n^{-\frac{1}{2}})$:*

$$
\left( \Psi(\tilde{P}) + \frac{1}{n}\sum_{i=1}^n \tilde{\phi}(O_i) \right) - \Psi(P) = O_p(n^{-\frac{1}{2}}).
$$

We now interpret the qualitative estimation implications of the analysis. Theorem 1 states the rate conditions required of *numerical approximation* in $\epsilon, \lambda$ to preserve the *statistical property* of rate

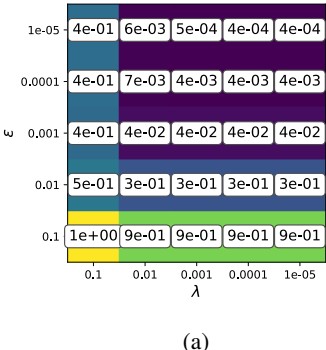
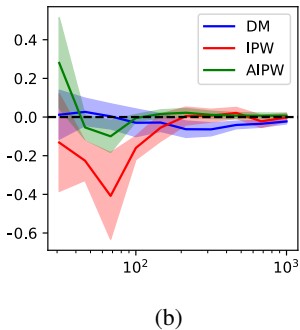

| (a) | (b) |

Figure 1: (a) Epsilon-lambda plot for mean potential outcome: MAE in approximation of one-step adjustment. (b) Error in estimation of $\mathbb{E}[Y(1)]$ over $n$.

double robustness. Assuming that the nuisances induced by the probability density estimates satisfy a product-rate condition that the product of their RMSEs is faster than the parametric $O(n^{-\frac{1}{2}})$ rate, the form of Proposition 1 suggests that the rate of RMSE convergence of perturbed nuisances is required to be faster than the rates of the unperturbed nuisance functions, $\max(r_\mu, r_e)$, which can be a slower rate than the $\epsilon = o(n^{-\frac{1}{2}})$ implied by the generic analysis of finite differences. For a direct comparison, the rate of Carone et al. [10, Thm. 5] suggests a rate requirement of $\left(\epsilon \lambda^{-d_1}\right)^s + \lambda^{-2} = o(n^{-\frac{1}{2}})$.

In general, the dimension dependence on $\lambda$ in Lemma 1 will prevent setting a slower rate for $\epsilon$ for nontrivial dimensionality. Note that without the smoothing requirement, the potential improvement in the rate of $\epsilon$ could be on the order of generic rate improvements implied by a central difference scheme. That is, we would have attained the approximation benefits of generically approximation-improved finite-difference schemes via the specialized *statistical* structure of the target adjustments.

# 5 Applications: Mean Potential Outcome and Dynamic Settings

As empirical illustrations, we first show a plot of numerical approximation error ranging over $(\epsilon, \lambda)$ in the simple case of the mean potential outcome. We then apply the tools that we derived in the previous section to analyze more complicated settings.

## 5.1 Mean potential outcome

We first conduct a validation of our decomposition and theoretical characterization. Our data-generating process includes a piecewise linear outcome model (such that kernel regression is mis-specified). We conduct plug-in evaluation of the mean potential outcome with kernel density esti-mates. In Figure 1a we include an $(\epsilon, \lambda)$-plot for the simple case of AIPW (see [10] for more discus-sion and examples). We consider a one-dimensional case with uniformly distributed $X$, piecewise-linear $Y$, and smooth propensity scores that are logistic in $\sin(X)$. We use $n = 500$ and fix the bandwidth $h = 0.05$. Colors denote magnitude of the mean absolute error, included in text on the heatmap. Without loss of generality, we study the estimation of a mean under missingness, $\mathbb{E}[Y(1)]$. Figure 1b illustrates the estimation error of various strategies with the comparable kernel-based es-timates (DM is regression adjustment). Note that even in this simple setting AIPW offers benefits and sample efficiency relative to DM. We include further numerical experiments in the Appendix.

## 5.2 Dynamic treatment regimes

We study the estimation properties of the empirical Gateaux derivative for multi-stage DTR from the probability density representation. In $T$-stage dynamic treatment regimes, the causal quantity of interest is the mean potential outcome $\mathbb{E}[Y(\bar{a})]$, where $\bar{a} = (a_0, \ldots, a_T)$ is the (deterministic) treat-ment strategy. Assuming $Y(\bar{a})$ is sequentially ignorable given the treatment and covariate history, $(\bar{A}_t, \bar{X}_t)$, at each time $t$, this causal quantity can be identified by the $g$-formula,

$$\mathbb{E}[Y(\bar{a})] = \int \mathbb{E}\left[Y \mid \bar{A} = \bar{a}, \bar{X} = \bar{x}\right] \prod_{t=1}^{T} \tilde{p}(x_t \mid \bar{a}_{t-1}, \bar{x}_{t-1}) \, d\bar{x},$$

where $\bar{A} = (A_0, \ldots, A_t), \bar{X} = (X_0, \ldots, X_t)$. To derive its influence function, we take the empirical Gateaux derivative by considering the perturbation in the direction of $o_i = (\bar{x}_i, \bar{a}_i, y_i)$, where $\bar{x}_i = (x_{i0}, \ldots, x_{it})$ and $\bar{a}_i = (a_{i0}, \ldots, a_{it})$,

$$\Psi(\tilde{P}^i_\epsilon) = \int \left( \int y \frac{\tilde{p}_\epsilon(y, \bar{A}=\bar{a}, \bar{X}=\bar{x})}{\tilde{p}_\epsilon(\bar{A}=\bar{a}, \bar{X}=\bar{x})} \, \mathrm{d}y \right) \prod_{t=1}^T \frac{\tilde{p}_\epsilon(x_t, \bar{a}_{t-1}, \bar{x}_{t-1})}{\tilde{p}_\epsilon(\bar{a}_{t-1}, \bar{x}_{t-1})} \tilde{p}_\epsilon(x_0) \, \mathrm{d}\bar{x}. \quad (4)$$

Below we characterize how this empirical Gateaux derivative at the smoothed distribution differs from the one at the (unsmoothed) estimated distribution. The following result is analogous to Proposition 1 and Corollary 1 but is extended to the dynamic treatment regime.

**Proposition 2.**

$$\epsilon^{-1}(\Psi(\tilde{P}^i_\epsilon) - \Psi(\tilde{P}))$$

$$= \left( \mathbb{E}_{\tilde{p}(\bar{x}_{1:T})} \left[ \mathbb{E}_{\tilde{P}_\epsilon} \left[ Y \mid \bar{A}, \bar{X} \right] \mid \bar{x}_0 \right] - \Psi(\tilde{P}) \right) + \frac{\mathbb{I}[\bar{A}=\bar{a}_i]}{\tilde{p}_\epsilon(\bar{a} \mid \bar{x})} \left[ Y_i - \mathbb{E}_{\tilde{p}(\bar{x})} \left[ \mathbb{E}_{\tilde{P}} \left[ Y \mid \bar{A}, \bar{X} \right] \right] \right]$$

$$+ \sum_{s=1}^T \frac{\mathbb{I}[\bar{A}=\bar{a}_{i,T-s}]}{\tilde{p}_\epsilon(\bar{a}_{T-s} \mid \bar{x}_{T-s})} \left\{ \mathbb{E}_{\tilde{P}} \left[ \mathbb{E}_{\tilde{P}_\epsilon} \left[ Y \mid \bar{A}, \bar{X} \right] \mid \bar{a}_{i,T-s}, \bar{x}_{i,T-s+1} \right] \right. \quad (5)$$

$$\left. - \mathbb{E}_{\tilde{P}} \left[ \mathbb{E}_{\tilde{P}_\epsilon} \left[ Y \mid \bar{A}, \bar{X} \right] \mid \bar{a}_{i,T-s}, \bar{x}_{i,T-s} \right] \right\} + O(\epsilon^2) + O(\lambda^\beta)$$

The expansion verifies that the requirements for numerical approximation are similar in $\epsilon$ as in the case of a single-timestep mean potential outcome, despite the additional complexity of the dynamic treatment regime function as a nested expectation. Again, smoothing incurs overall unfavorable dependence in the dimension, as the extension of Lemma 1 to $\mathbb{E}_{\tilde{P}_\epsilon}[Y \mid \bar{A}, \bar{X}]$ also incurs unfavorable dependence on the dimension. The history-dependent nuisances imply that the dimension additionally grows in the time horizon.

## 5.3 Perturbations for (constrained) infinite-horizon off-policy evaluation

We show how this framework can be applied to derive generic bias corrections for infinite-horizon off-policy optimization in offline reinforcement learning, via the canonical *linear programming* characterization [17, 61]. Our application is relevant to a recent line of work on off-policy evaluation and learning (OPE/L) [5, 43, 47–49, 67, 68].

**Setup.** We recall the classical linear programming formulation of finding an optimal policy in a tabular (finite-state, finite-action) Markov decision process, due to De Farias and Van Roy [17], Puterman [61]. Recent work has revisited this formulation with interest in developing primal-dual algorithms [1, 55, 65, 74]. The dynamics follow an infinite-horizon Markov decision process with discount factor $\gamma$. The offline dataset is comprised of trajectories of state, action, next state observations: $\{(s_i^0, a_i^0, \ldots, s_i^t, \ldots, s_i^T)\}_{1:N}$. We derive our bias adjustment under a statistical model where the stationary distribution factorizes as $p(s, a, s') = d(s, a)P(s \mid s, a)$. From the joint state-action-state occupancy distribution $p(s, a, s')$, we estimate the marginalization $d(s, a)$, the stationary state-action occupancy distribution, which we use to estimate the transition probability $P(s' \mid a, s) = \frac{p(s, a, s')}{d(s, a)}$. Let $\mu_0(s)$ denote the initial state distribution, estimated from offline data and assumed to be equivalent in the original confounded and evaluation setting. Because we focus on the tabular case with discrete distributions, we do not require smoothing.

**The optimal policy linear program.** The *optimal* value function solves the following linear program in a finite-state and finite-action setting; the objective value at optimality is the policy value. Here, $P_a \in \mathbb{R}^{|S| \times |S|}$ is transition matrix at action $a$, with $P_a(s, s') = P(s' \mid s, a)$, $\mu_0(s)$ is the marginal occupancy distribution of the initial state. We assume, as is fairly common in theoretical analyses of reinforcement learning, that the reward function is a known function of state and action, $r(s, a)$. Let $r_a = \{r(s, a)\}_{s \in \mathcal{S}}$ be the vector of reward values per state, for fixed action $a$. We have:

$$\Psi_D(P) = \min_V \{(1 - \gamma)\mu_0^\top V : (I - \gamma P_a) V - r_a \geq 0, \quad \forall a \in \mathcal{A}\} \quad (6)$$

The dual formulation is well known to parametrize the stationary occupancy probabilities of the optimal policy (at optimality):

$$\Psi_P(P) = \max_\mu \left\{ \sum_{a \in \mathcal{A}} \mu_a^\top r_a : \sum_{a \in \mathcal{A}} (I - \gamma P_a^\top)\mu_a = (1 - \gamma)\mu_0, \quad \mu_a \geq 0, \quad \forall a \in \mathcal{A} \right\}. \quad (7)$$

We verify the bias correction from the Gateaux derivative $\frac{\mathrm{d}}{\mathrm{d}\epsilon}\Psi_D(P_\epsilon)\Big|_{\epsilon=0}$. Note that evaluating this functional at the perturbed distribution evaluates the linear program with perturbed left-hand side coefficients; i.e., perturbing the constraints of the coefficient matrix as well as the objective (marginalizing over a perturbed initial state distribution. We evaluate the Gateaux derivative in the direction of $(\tilde{s}, \tilde{a}, \tilde{s}')$. We assume nondegeneracy (unique optimal solution); this can be relaxed.

**Assumption 2.** *The optimal solution is unique; there is no degeneracy.*

**Proposition 3.** *Perturb in the direction of a generic observation $(\tilde{s}, \tilde{a}, \tilde{s}')$ :*

$$\frac{\mathrm{d}}{\mathrm{d}\epsilon}\Psi_D(P_\epsilon)\Big|_{\epsilon=0} = (1-\gamma)V^*(\tilde{s}) - \frac{\mu^*(\tilde{s}, \tilde{a})}{d(\tilde{s}, \tilde{a})}\left(r(\tilde{s}, \tilde{a}) + \gamma V^*(\tilde{s}') - V^*(\tilde{s})\right) - \Psi_D(P) \quad (8)$$

**Sketch of argument.** The statement of Proposition 3 is not surprising since we can immediately verify the mean-zero sample version of the Bellman residual, and so one might plausibly derive this form from double robustness. Rather, we provide an alternative argument based on the sensitivity analysis characterization of [25].For $\epsilon$ small enough, the *active basis* remains the same. When the perturbation matrix is $P + \epsilon G$ for some matrix $G$, [25] notes that the derivative of the optimal value is $d'(\epsilon) = -\bar{\pi}G\bar{x}$ (where $(\bar{\pi}, \bar{x})$ are the dual and primal optimal solutions, respectively) can be obtained from the derivative of a matrix inverse, which yields higher-order expansions.

We next compute the approximation error that arises from evaluating finite differences, so long as $\epsilon$ is small enough to maintain the same active basis, which is empirically verifiable.

**Proposition 4** (Error analysis from finite differencing). *Perturb in the direction of a generic observation $(\tilde{s}, \tilde{a}, \tilde{s}')$. Then:*

$$\epsilon^{-1}(\Psi(P_\epsilon) - \Psi(P)) = (1-\gamma)V_\epsilon^*(\tilde{s}) - \frac{\mu^*(\tilde{s}, \tilde{a})}{d_\epsilon(\tilde{s}, \tilde{a})}\left(r(\tilde{s}, \tilde{a}) + \gamma V(\tilde{s}') - V_\epsilon(\tilde{s})\right) - \Psi_D(P) + O(\epsilon) \quad (9)$$

In this setting, unlike the earlier case of the treatment mean in Corollary 1 and Theorem 1, even in a favorable discrete-state case without smoothing requirements, rate double robustness does not admit weaker numerical requirements on $\epsilon$. In the analysis, eq. (45) shows that $V_\epsilon(s) - V(s) = O(\epsilon)$, so that Proposition 4 holds with the unperturbed nuisances and the same order of approximation error.

**Generalization to constraints.** We argue the main benefit of this approach is that by computing these finite differences, we can obtain generic bias corrections via a more broadly applicable argument that does not require interpretation of the closed-form solution to an optimization problem. For example, consider a relevant subclass of constrained Markov decision processes comprised of additional constraints on the policy variables: the linear programming formulation is particularly appealing because linear constraints can be directly added [3]. The additional constraint $\mu \in \mathcal{P}$ reflects linearly representable constraints on state-occupancy under the optimal policy. For example, relevant constraints could require the occupancy in certain risky states to be bounded. [59] develops a framework with a "caution function," a convex function on the state-action occupancy measures.

**Example 3** (Constrained policy optimization).

$$\Psi_P(P) = \max\left\{\sum_{a\in\mathcal{A}}\mu_a^\top r_a : \sum_{a\in\mathcal{A}}(I - \gamma P_a^\top)\mu_a = (1-\gamma)\mu_0, \ \mu_a \geq 0, \ \forall a \in \mathcal{A}, \ \mu \in \mathcal{P}\right\}.$$

# 6    Conclusions

We have presented a constructive algorithm that approximates Gateaux derivatives finite differencing, with a focus on the statistical functionals used in causal inference. There are several directions which are natural follow-ups to this work. First, the smoothing perturbation, although it restores absolute continuity, introduces technical barriers in the implied approximation rates with regards to dimension dependence. Another useful direction would be to explore methods that are fully adaptive, not requiring any prior knowledge of beneficial statistical properties. Finally, our work has been mainly theoretical, and it would be useful to conduct further empirical comparisons.

**Acknowledgements.** Angela Zhou gratefully acknowledges support from the Foundations of Data Science Institute and the Simons Institute's Program on Causality. Part of this work was done while the author was visiting the Simons Institute for the Theory of Computing. Yixin Wang acknowledges National Science Foundation Grant NSF-CHE-2231174. Michael Jordan acknowledges support from the Vannevar Bush Faculty Fellowship program under grant number N00014-21-1-2941.

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
