# OpenReview forum: "Empirical Gateaux Derivatives for Causal Inference"
_NeurIPS.cc/2022/Conference — NeurIPS 2022 Accept_

### Official Review · Reviewer_54Sg · 2022-07-11

**Rating:** 6
**Confidence:** 4
**Soundness:** 3 good
**Presentation:** 2 fair
**Contribution:** 2 fair

**Summary:**

Efficient / doubly-robust estimation in causal inference typically requires constructing estimators based on the (efficient) influence function, which is derived case by case. The paper considers numerical approximation to the influence function with *empirical Gateaux derivatives*, which are obtained through finite differencing and smoothed perturbation. The approximation uses plugin estimate of the probability density. The paper studies conditions on the smoothing and perturbation parameter such that the resulting one-step estimator retains desired statistical properties. The paper first examines the case of estimating the counterfactual mean and then extends to more complex dynamic settings.

**Questions:**

1. Gateaux differentiable vs influence function

(1) The definition of the influence function ($\S$2) in terms of Gateaux derivative comes from the robust statistics literature. How is it related to the influence function used for semiparametric inference?

(2) page 2, line 59-60: in the direction of $H$ or for all $H$?

2. On plugin estimation through density estimation

(1) Given the curse of dimensionality, it is unclear to me why the approximation must be based on a density estimate. I do not parse the argument provided in line 117-121: can the authors elaborate on this? More specifically, for the running example, it seems that one can easily simulate data from the perturbed distribution. Can the approximation be based on an estimator using the simulated perturbed data (e.g., empirical average of an estimated conditional mean, where the distribution is that of the simulated data)?

(2) Suppose density estimation is necessary. Then it is worth discussing when the kernel-based estimators can fulfill the conditions required, e.g., (iv) in Assumption 1.

(3) To really "bolster confidence in the numerical procedure", the case study of the running example should be expanded accordingly to show that the proposed method works in a range of settings (e.g., at least when dimension is not high).

3. I have a hard time trying to follow the extension to dynamic settings in $\S$4. Can the authors summarize the take-away message there?

4. A few points worth clarification.

(1) How is $\tilde{P}$ used in Algorithm 1?

(2) line 71: it is unclear what "$\Psi(P_{\epsilon,\lambda}^{\ast})$ is the projection of $\Psi(P_{\epsilon,\lambda})$ ..." means.

5. On the relation to previous works in “automatic” or “computerized” semiparametric inference

The paper features a long list of references. Since the focus here is the influence function, I would suggest adding a paragraph after $\S$2, which clarifies the relation to previous papers on "computerizing" influence-function based estimations, e.g., [9] and [17] in the bibliography.

**Limitations:**

I do not foresee a potential negative societal impact of the proposed work.

**Strengths And Weaknesses:**

### Strengths
1. The paper is built upon recent efforts in "automating / computerizing" influence-function-based estimation and studies an important question.
2. The paper provides a clean analysis for the basic case of estimating the counterfactual mean.

### Weaknesses
1. The approximation is based on a density estimator of the underlying data distribution, which seems to bear serious practical limitations. It is unclear if this choice is necessary, and if so, how to overcome the limitations.
2. The conditions impose a product-form rate condition on the smoothed estimates; see (iv) of Assumption 1. It is unclear when these conditions can be met in practice.
3. Other than a simple simulation, the paper does not seem to provide concrete practical recipes for the running example that can flexibly incorporate different underlying distributions, dimensions, smoothness, etc. This weakens the relevance for the results developed for the more complex, dynamic settings.
4. The clarity of certain sections can be improved.

---

> ### Author Response · Authors · 2022-08-02
> **response to weaknesses**
>
> - Weakness 1: ```density estimator ... practical limitations```
>
>  In the main text we have focused on Kernel density estimation/Nadaraya-Watson regression because it is a classical “textbook” example of the kind of “plug-in nuisance”, i.e. a conditional expectation estimator arising from integration of a density estimator (or in machine learning vocabulary, a generative model). Please note that in our appendix lines 676-688 we discuss more complex generative models/plug-in nuisance estimations that follow a similar paradigm, and are amenable to scaling up the procedure.
>
> - Weakness 2: ``` product-rate conditions ...```
>
> We separately analyze the bias due to smoothing from the estimation error of the induced nuisances (without smoothing). This allows us to appeal to the same types of conditions of product-rate assumptions, which are standard in double/orthogonalized machine learning, as are assumed in that literature. That large literature has the same issue with nonparametric curse of dimensionality: hence primitive conditions are given to satisfy the product-rate conditions under low-dimensional structure, i.e. sparsity, intrinsic dimension, etc. We will add a paragraph to overview these results. For example, Section D of Foster and Syrgkanis' Orthogonal Statistical Learning collects some results on empirical risk minimization and kernel estimation and rates.
> In general, of course some sort of lower-dimensional structure will have to be assumed, such as intrinsic dimension (as in Khosravi et al.'s Non-parametric inference adaptive to intrinsic dimension, which studies trees, whose neighborhood weights admit kernels that can be used as linear smoothers), weight matrix norm bounds/parameter counts for neural networks, etc. The restrictiveness of the product-rate assumption is no worse in our paper than in the large literature on double/orthogonalized machine learning. We will also include a paragraph referring to these other papers and rates.
>
> - Weakness 3: ``` no concrete practical recipes for the running example ... flexibly incorporate different underlying distributions, dimensions, smoothness```
>
> We disagree. We summarize the concrete, practical recipe in Algorithm 1. Simply plug in your favorite distribution/density estimate, using whichever inductive biases you prefer, to conduct plug-in evaluation of the statistical functional. We will clarify this. Again, in line 126 we said, "Of course, other estimates can be used: [see] Appendix E for discussion of other choices of density estimates." and in Appendix E we included examples of other density estimates that are more "modern", eg. nonparametric mixture models, etc. These can be equivalently plugged in, but we do not contribute along these lines, so we didn't include this in the main text due to space limitations. Our analysis is modular so that we analyze the bias induced by smoothing the evaluation point in an additive way. We will further clarify the modularity of the analysis.

---

> ### Author Response · Authors · 2022-08-02
> **response to 54Sg questions (1/2)**
>
> Thanks for the feedback, we hope our response clarifies the suggested weaknesses and points to some discussions in the main text. We appreciate the suggestion to have a self-contained discussion of satisfying assumption (iv) and will include a paragraph along these lines, even if that paragraph mostly follows the rest of the works on double/orthogonalized machine learning on satisfying product error rates, because of modularity of our analysis.
>
> We respond to specific questions below:
>
> # 1
>
> ### 1.1
>
> For use in causal inference/semiparametric statistics, the influence function definition is used with a specific choice of the perturbation distribution H, namely a dirac Delta distribution at the observation. This is then used in the one-step estimator. Semiparametric inference further requires smoothing, so that paths are absolutely continuous under the (semiparametric) model.
> ### 1.2
>
> For all H – we will clarify.
>
> # 2
>
> ### 2.1.
>
> We argue the approximation must be based on a density estimate for more general applicability of the “automatic approach”. More concretely, current derivations would compute the analytic derivative of $ \frac{d}{d\epsilon} \int y \frac{p_{\epsilon}(y, A=1, x)}{p_{\epsilon}(A=1, x)} p_{\epsilon}(x) d y d x$, and then simply the analytical derivative, for example recognizing that $\int y  \frac{p(y, A=1, x)}{p(A=1, x)} dy = E[Y\mid A=1,X=x]$. Then the analytical derivative may choose any estimator of  $E[Y\mid A=1,X=x]$ to plug into the form of the influence function. But this requires analytical simplification/appropriate interchanges of integrals and differentiation. A truly automatic approach cannot make these simplifications. An alternative could maybe be to use graphical criteria to identify the appropriate marginalizations, but we don’t explore those approaches.
>
> ``` Can the approximation be based on an estimator using the simulated perturbed data (e.g., empirical average of an estimated conditional mean, where the distribution is that of the simulated data)?```
>
> We understand this proposal to suggest to use the empirical distribution as the estimator of the density, and evaluate the perturbed distribution by perturbing the empirical distribution. Our characterization shows that this would correspond to using the empirical distribution as estimates of what are ultimately conditional densities, which would lead to induced nuisance estimators that do not have favorable convergence properties. Although this could work for something like “automatic” influence function for the mean, for the covariate-conditional regression-adjustment identification in causal inference, the empirical distribution is not a good estimator of conditional densities.
>
> ### 2.2 satisfying product-error rate requirements
>
> We will expand on this discussion. Our rates suggest the product rate conditions can be satisfied there is sufficient smoothness of the density relative to the dimension.
>
> ### 2.3
>
> The focus of this paper is on clean theoretical characterizations which facilitate understanding of what is happening “under the hood” of the numerical approximation approach. (Previous work does provide some numerical evidence, but not this analytical characterization). We agree this is an important direction for future work. We found implementation details (integration method, etc) to matter in our experiments which would muddy the comparisons. We will highlight this as a direction for future work.
>
> # 3. take-away messages of DTR examples
>
> Good point, we will add some exposition here to have this be more self-contained, perhaps for an unfamiliar audience.
>
> Takeaway messages:
>
> - The automatic approach can be useful in the challenging setting of a DTR, where it is not obvious what the appropriate nuisance functions would be. For example, the takeaway is that one could evaluate an empirical Gateaux derivative using “model-based approaches” in (offline) reinforcement learning that plug in some estimate of the transition distributions. These “model-based approaches” are common, and one-step adjustments could be computed by simply perturbing the transition density estimates that are already being used.
>
> - The second take-away message is that estimators that build on the classical linear programming approach admit influence functions that can be computed by perturbing the transition probability estimate inputs. Our argument/proof is novel and directly extends to settings with custom constraints. Other approaches required knowledge of the optimal dual variables: changing the constraints changes the interpretation of the dual variables, making it harder to deduce the form of the influence function. In contrast, we show that the finite-differences/perturbation argument can be directly extended for custom constraints.

---

> ### Author Response · Authors · 2022-08-02
> **response to 54sg questions (2/2)**
>
> # 4
>
> ### 4.1
>
> Typo, thanks for noting. We will change assumed input to be $P$
>
> ### 4.2
>
> We included this discussion for completeness, but since our paper focuses on nonparametric models we will remove from the main text to prevent further confusion. It means projection of the distributions onto the semiparametric model; for example for DTRs this can include additional conditional independences, etc.
>
> #5. relationship to related work
>
> We will clarify this. Re: [9,17], see lines 306-311 for direct comparison. And lines 61-75 provide a thorough comparison to [9].

---

> > ### Comment · Reviewer_54Sg · 2022-08-08
> > **Thanks**
> >
> > I would like to thank the authors for their detailed reply.
> >
> > I have a few more comments to follow up.
> >
> > ### On 1.1
> >
> > > For use in causal inference/semiparametric statistics, the influence function definition is used with a specific choice of the perturbation distribution H, namely a dirac Delta distribution at the observation.
> >
> > While this might be a practical strategy to derive the influence function, I would not take this as a definition. In general, I think the influence function employed in asymptotic linear expansion (e.g., for one-step estimation) would require a uniform sense of differentiability; see, e.g., van der Vaart ($\S$20.2, 2000). This is a point perhaps worth clarification: Gateaux differentiability does not guarantee valid semiparametric inference with the influence function.
> >
> > ### On 2.1
> >
> > > But this requires analytical simplification/appropriate interchanges of integrals and differentiation. A truly automatic approach cannot make these simplifications.
> >
> > I am not sure about this. Supposedly the simplification can be performed by some computer algebra program (assuming sufficient regularity).
> >
> > ### On 2.2
> >
> > Could you remind me when this condition can be satisfied with the kernel estimates suggested in the paper?

---

> > > ### Author Response · Authors · 2022-08-08
> > > **reply**
> > >
> > > Thanks for reading and for your questions. We clarify below:
> > >
> > > ## on 1.1
> > >
> > > ```This is a point perhaps worth clarification: Gateaux differentiability does not guarantee valid semiparametric inference with the influence function.```
> > >
> > > We agree, and in our response to you we more closely conflated the terms “causal inference/semiparametric statistics” than we do in the paper.
> > >
> > > In our paper, we are careful to make this separation distinct; see Remark 1 describing how we focus on influence function adjustments rather than Semiparametric inference. It is also true that the one-step estimator is justified under asymptotic linearity, which is a stronger condition than supposed in the definition, as we discuss in lines 93-94. We intentionally separately discussed the influence function and asymptotic linearity. We agree it may not be obvious from the abridged discussion in the main text that asymptotic linearity is a stronger requirement of the functional. We propose to add a line emphasizing this explicitly at line 93.
> > >
> > > We will further clarify between: the presentation of “influence curve in the sense of Hampel/Huber” as the limit, when it exists, $\lim_{\epsilon \to 0} \epsilon^{-1}\left(\Psi\left(P_{\epsilon, \lambda}\right)-\Psi(P)\right)$ (uniformly over points of evaluation of the delta function) and /separately/ include the definition of Gateaux derivative in the sense of van der Vaart 2000, 20.2. When the functional is Gateaux-differentiable, its definition can be instantiated with the Dirac delta function to recover the former.
> > >
> > >  We think explicitly adding these distinctions will help avoid confusion, thanks for pointing out a source of potential confusion.
> > >
> > > ## on 2.1
> > >
> > > To clarify, what we mean is that some of the referenced approaches proposing computerized approaches exactly used some analytical prior knowledge to identify the right nuisance functions, without the use of computer algebra to repeat the exercise for more general functionals; namely Frangakis et al. and Escanciano et al study the counterfactual mean with a conditional expectation nuisance.
> > >
> > > We agree that these simplifications, for cases such as the counterfactual mean and longitudinal/mediation-type analyses, do seem like opportunities for symbolic computation, more rule-based approaches, and/or graphical algorithms. If the reviewer has a specific paper in mind, we would be happy to include a reference to that also. In general we feel that this is also a good direction for future work.
> > >
> > > We will make this explicit to avoid potential confusion.
> > >
> > > ## on 2.2
> > >
> > > The conditions are satisfied when $$(\epsilon^{2} \lambda^{-d} + \lambda^\beta) = o(n^{-\frac 14})$, for smooth kernels in a Holder class (first term), and the induced (unperturbed) nuisances also satisfy the rate implied by the product rate condition (second term).
> > >
> > > This first term imposes conditions on $\lambda$, which is a parameter the analyst may choose. This dependence imposes that $\epsilon \ll \lambda$.
> > >
> > > Satisfying conditions on the second term is equivalent to assuming the induced, unperturbed nuisances satisfy product rate conditions. (For example, it’s sufficient to assume both the first term and the second term are $o(n^{-\frac 14})$.
> > >
> > > The minimax rates for nonparametric density estimation are MSE order $O(n^{-2 \beta /(2 \beta+d)})$, for densities in a Holder class of order $\beta$ (these are indeed achieved by the kernel density estimates we study for optimal choices of bandwidth). The most common case is $\beta=2$, in which case the minimax rate suggests in a univariate case $O(n^{-4/5})$. Indeed, product rate conditions help in this setting. Of course, this rate exhibits the nonparametric curse of dimensionality: additional structure (such as higher-order smoothness, i.e. assumptions of being in a Holder class of higher order) are required.
> > >
> > >  We will add this explication: in the main text, we can probably only fit a reminder of the minimax rate for nonparametric estimation and the first paragraph; we will elaborate in the appendix.

---

> > > > ### Comment · Reviewer_54Sg · 2022-08-09
> > > > **Thanks**
> > > >
> > > > Thanks for the clarifications. They are very helpful!

---

### Official Review · Reviewer_cYXd · 2022-07-11

**Rating:** 7
**Confidence:** 3
**Soundness:** 3 good
**Presentation:** 3 good
**Contribution:** 3 good

**Summary:**

Gateaux derivatives arise in a number of statistical estimation problems, and are used to improve asymptotic efficiency in estimation (e.g., in one-step estimators).  For a given problem, the form of the Gateaux derivative is typically derived analytically, and the relevant components then estimated from data (e.g., certain conditional expectations).  Here, the form of the derivative is known, and typical analysis focuses on how different rates of estimation (in the relevant nuisance parameters) lead to appropriate rates of convergence in the original problem.

This paper considers "empirical" Gateaux derivatives, which do not involve such analytic derivations, but instead directly seek to estimate the derivative via finite-differencing.  This introduces an additional source of error in computing the Gateaux derivatives (numerical approximation error), but has the potential to broader the utility of one-step estimators, by not requiring explicit derivations in the case of new statistical functionals.

While other work has considered the problem of numerical approximation error, this work analyzes how both numerical and statistical errors interact in the context of several examples: One of the main results, for instance, gives rates on numerical approximation hyperparameters that preserve parametric rates in the one-step estimator for a standard causal inference problem (counterfactual mean estimation).  Two other illustrative estimation problems are considered as well.


**Questions:**

My questions can be found in the "Strengths and Weaknesses" section, where every bullet point corresponds to a question.  I collect them (verbatim) below.

* Line 188: "can be a slower rate than implied by the generic analysis of finite differences".  What kind of rate would that be, and is there a reference for such results?
* Lines 190-191: "potential improvement...could be on the order of generic rate improvements implied by a central difference scheme".  What is this referring to?
* Lines 278-280: "does not appear that rate-double robustness would admit weaker numerical requirements on $\epsilon$".  Weaker requirements than what?  The "generic analysis" referenced above?
* Are there "conservative" rates on $(\epsilon, \lambda)$ that will always preserve $O_P(n^{-1/2})$ rates of estimation, obtainable via some generic analysis?
* Proposition 2 (DTR) "verifies that the requirements...are similar in $\epsilon$ as in the case of a single-timestep", but there is no $O(\epsilon^2)$ term in either Proposition 1 or Corollary 1.  Could you clarify what is meant here?
* Propositions 3 & 4 differ not only in an additive term, but also in the usage of perturbed nuisances, which makes them difficult to compare directly (this also applies to Corollary 1, as noted on line 170).  Is there a reason why a direct comparison (e.g., isolating only an additive difference) is unnecessary here?
* Are there existing scenarios where the analytic form of the gateaux derivative is non-obvious, but where these conditions (pathwise differentiability, second-order remainder) can nonetheless be verified to hold? Or does verifying these conditions *always* require derivation of the analytic form?
* More broadly, are there scenarios where we can apply this approach (with appropriately conservative rates on $(\epsilon, \lambda)$) and have confidence in achieving $O_P(n^{-1/2})$ rates, **without** deriving the analytic derivative?  E.g., would the constrained MDP with arbitrary linear constraints be such an example?
* In the display following line 563, it is claimed that the following holds due to Cauchy-Schwarz.  I'm not sure I see the application of CS here: Is there another reason why we would expect the cross term $2 E[(\tilde{\mu}_{\epsilon}(X) - \tilde{\mu}(X))(\tilde{\mu}(X) - \mu(X))]$ to be non-positive?

$$ E(\tilde{\mu}\_{\epsilon(X)} - \mu(X))^2 \leq E(\tilde{\mu}_{\epsilon}(X) - \tilde{\mu}(X))^2 + E(\tilde{\mu}(X) - \mu(X))^2 $$
* In the display following line 562, the last inequality seems like a non-trivial jump, would you mind walking through the logic explicitly?


**Limitations:**

I think the authors do a fine job of explaining limitations of the approach.

**Strengths And Weaknesses:**

[**REBUTTAL UPDATE**: See discussion below.  My main concerns were addressed during the response, and I am convinced that the relevant clarifications are straightforward to incorporate into a camera-ready, if accepted.  My score has been updated as follows:
* Presentation: 2 -> 3
* Overall Score: 5 (borderline accept) -> 7 (accept)
]

### Strengths
This paper tackles an interesting and highly relevant problem, and provides novel technical results. Theorem 1 in particular is a clean and informative result, characterizing rates of perturbation and smoothing that preserve $\sqrt{n}$-consistency of counterfactual mean estimation with empirical Gateaux derivatives.  Overall, one of the main take-away messages is that the structure of the statistical problem can lead to weaker approximation requirements (e.g., rates on $(\epsilon, \lambda)$) in some cases, but not in others, which I thought was an interesting point.

### Weaknesses

The clarity and quality of presentation could be improved, which is a weakness that is hopefully straightforward to fix.  Below, I will detail some points that were particularly unclear to me, in the hope that it is useful for the authors as they craft a response. I put explicit questions in "bullet points", with the remaining text as context.  I am willing to increase my score if these points can be sufficiently clarified.

**(1) Unclear significance of analysis of required rates for $(\epsilon, \lambda)$**

As I understand it, the main goal of the analysis is to understand the extra error incurred by using a finite-differencing approach, above and beyond using analytic derivatives:  As in Theorem 1, if this error decays fast enough, then one can use empirical derivatives while preserving $O_p(n^{-1/2})$ rates of estimation.  This seems like a very clear type of result, but for an application where the analytic derivative is already well known.  Meanwhile, the results in the remaining sections do not go "all the way" to a result like Theorem 1, but instead stop at giving some side-by-side comparison of the analytic and empirical derivatives.

It seems that there are two conclusions the reader is supposed to draw from this analysis, in the context of the remainder of the work:  First, that $(\epsilon, \lambda)$ can decay slower than we might generically "expect", for problems with special structure. Second, that this special structure is present in the dynamic treatment regime (DTR) functional, but not in the policy optimization functional.  This second point is implied to be surprising, because all three problems exhibit some double-robustness structure (see lines 279-280).

I had trouble drawing such conclusions, though I suspect this is mostly an issue of presentation / clarity.

(1a) First, it seems in several places that the reader should have a "baseline" result in mind, to contrast with the results presented here, but this baseline was not entirely clear.  A few examples:
+ Line 188: "can be a slower rate than implied by the generic analysis of finite differences".  What kind of rate would that be, and is there a reference for such results?
+ Lines 190-191: "potential improvement...could be on the order of generic rate improvements implied by a central difference scheme".  What is this referring to?
+ Lines 278-280: "does not appear that rate-double robustness would admit weaker numerical requirements on $\epsilon$".  Weaker requirements than what?  The "generic analysis" referenced above?
+ Are there "conservative" rates on $(\epsilon, \lambda)$ that will always preserve $O_P(n^{-1/2})$ rates of estimation, obtainable via some generic analysis?

The first three questions are about contextualizing the results, but the last one this is important for clarifying whether (a) there is always a generic approach to derive rates on $(\epsilon, \lambda)$ that preserve $O_P(n^{-1/2})$ convergence, and this is just an improved analysis for specific estimands that shows slower rates are possible, or (b) it is generally necessary to do a "Theorem 1-style" analysis to verify $O_P(n^{-1/2})$ convergence.  The latter conclusion seems much more restrictive than the former.

(1b) Second, conclusions are often made by comparing the form of the empirical and analytical derivatives directly, but these were somewhat difficult to follow:
* Proposition 2 (DTR) "verifies that the requirements...are similar in $\epsilon$ as in the case of a single-timestep", but there is no $O(\epsilon^2)$ term in either Proposition 1 or Corollary 1.  Could you clarify what is meant here?
* Propositions 3 & 4 differ not only in an additive term, but also in the usage of perturbed nuisances, which makes them difficult to compare directly (this also applies to Corollary 1, as noted on line 170).  Is there a reason why a direct comparison (e.g., isolating only an additive difference) is unnecessary here?

**(2) Unclear significance of limitations of Empirical Gateaux derivatives**:

As outlined in the introduction, constructive / algorithmic approaches to bias adjustment are very appealing, particularly for problems where small changes require re-derivation of the analytic derivative. This paper strikes an appropriate note of humility in the conclusion, giving limitations of Empirical Gateaux derivatives as a "completely general approach" (lines 329-335), namely the fact that (a) pathwise differentiability and (b) the second-order nature of the remainder must be verified analytically. However, these limitations do seem to undercut the general value of the approach. With that in mind, a few relevant questions
* Are there existing scenarios where the analytic form of the gateaux derivative is non-obvious, but where these conditions (pathwise differentiability, second-order remainder) can nonetheless be verified to hold? Or does verifying these conditions *always* require derivation of the analytic form?
* More broadly, are there scenarios where we can apply this approach (with appropriately conservative rates on $(\epsilon, \lambda)$) and have confidence in achieving $O_P(n^{-1/2})$ rates, **without** deriving the analytic derivative?  E.g., would the constrained MDP with arbitrary linear constraints be such an example?

If there exist some space of problems where the answer to these questions is "yes", then it would go a long way towards mitigating the impact of these limitations.

**Comments on soundness**: Regarding technical soundness, I have a (hopefully minor) question or two on Lemma 1
* In the display following line 563, it is claimed that the following holds due to Cauchy-Schwarz.  I'm not sure I see the application of CS here: Is there another reason why we would expect the cross term $2 E[(\tilde{\mu}_{\epsilon}(X) - \tilde{\mu}(X))(\tilde{\mu}(X) - \mu(X))]$ to be non-positive?

$$ E(\tilde{\mu}\_{\epsilon(X)} - \mu(X))^2 \leq E(\tilde{\mu}_{\epsilon}(X) - \tilde{\mu}(X))^2 + E(\tilde{\mu}(X) - \mu(X))^2 $$
* In the display following line 562, the last inequality seems like a non-trivial jump, would you mind walking through the logic explicitly?

Otherwise, the proofs seem correct to me.  Note that I only read through the proofs for Section 3 in depth, and only skimmed the proofs of other relevant results (e.g., Propositions 3 and 4).  While I am well-versed in the causal inference literature, I am not otherwise an expert on non-parametric / semi-parametric statistics, so I may have missed something.

As an aside, it may be helpful to include a citation in the proof for some of the assumed results regarding kernels. [58] is referenced in the main text, referring to kernel smoothing more broadly, but it seems some of the prerequisite results could be cited more precisely (e.g., Lemma 25.1 of [58] appears to be a relevant result)

### Other Minor Feedback

I consider the following points to be minor feedback re: presentation / notation / possible typos, and they did not meaningfully influence my score, and they **do not require an explicit response from the authors** (some are stated as questions only because I am unsure if they are typos).

**Suggestions on clarity**:
* (Lines 15-19) This and some other sentences are a bit long and difficult to parse, and could perhaps be split into multiple sentences.
* (Line 71) Is the introduction of projections onto the semi-parametric model necessary, given Remark 1's statement that this work focuses on nonparametric models?  CLvdL (Equation 2.2) seems to refers to Luedtke, Carone, and van der Laan (2015) as a reference for the equation on between lines 71-72 holding generally in a non-parametric model.

**Other typos / inconsistencies**
* Example 1 uses $\mathbb{E}_P$ for the outer expectation, but not for the inner expectation.
* Proposition 1 uses $\tilde{\mathbb{E}}_{\tilde{P}_\epsilon}$ in one place, perhaps the tilde on $\mathbb{E}$ was not intended?
* Line 65, what is the observation $\tilde{o}$?  This is not referenced anywhere, I assume this is meant to be $o'$.
* Footnote 1, should the kernel be $\lambda^{-d} K(u / \lambda)$ instead of $h^{-d} K(u / \lambda)$?  There is also a reference to an $O(h^J)$ error term on line 568 that should perhaps be $O(\lambda^{\beta})$?
* Algorithm 1: Should it be $\tilde{P}$ on lines 3-4?  Should it likewise be $P_{\epsilon, \lambda}^i$ instead of $P_{\epsilon}^i$?
* Lemma 1: Should $\tilde{e}\_{\epsilon}(X) = \tilde{p}\_{\epsilon}(A = 1, X) / \tilde{p}(X)$ instead of the current formulation, which uses $\tilde{p}_{\epsilon}(A = 1 \mid X)$ in the numerator?  This would also make it consistent with usage in the proof (see e.g., line 561)
* Line 154, there seems to be a missing parenthesis
* Line 149 "analyses of from kernel density estimation"
* Assumption 1 (iv), should the equation refer to $\tilde{\mu}_{\epsilon}$ or just $\tilde{\mu}$?  Additionally, should the product-rate condition be $o_p(n^{-1})$ as written or $o_p(n^{-1/2})$?
* Line 561, says to bound perturbed $e$ one should "argue similarly", seemingly in reference to the (later) bound on the perturbed $\mu$, perhaps the order was swapped.
* Line 562, following equation, third equality, missing a square on the final $\tilde{p}(A=1,x)$ term.
* Line 572, $E[\Gamma(O; e_{\epsilon}, \mu)]$ should be $E[\Gamma(O; \tilde{e}_{\epsilon}, \tilde{\mu})]$
* Line 637, indicator is missing an $\epsilon$ on the left-hand side
* Supplement, Section D.2, the reference is to citation [20], but I believe this should be citation [18]

**Undefined notation**:
* Line 66, I did not see a definition of the function $g(u)$.
* I'm unsure if the dimension $d$ was precisely defined prior to usage in Lemma 1, though it is fairly obvious from context.
* Eq. 7: I'm not sure if $\mu_a$ is defined anywhere, aside from being the optimization variable, and similar for $\mu^*(s, a)$ in Equation 8.
* $\nu$ is used a few times in the proofs without being defined (end of equation starting on 576, end of equation starting on 562), presumably referring to a strong-overlap constant.

---

> ### Author Response · Authors · 2022-08-02
> **response to reviewer cYXd, 1)**
>
> Thanks for the feedback and your careful and thoughtful response. We would first like to respond to your major concerns regarding weaknesses. We are confident that this can be straightforwardly addressed by more inline discussion of results in the previous literature that may not be widely known. Our submission was narrowly bracketed to be appropriately humble (ie. not overclaim novelty), but given that some of the results in prior literature may not be widely known, we propose to use the extra page in a camera-ready version to discuss inline/re-state these relevant results from prior work.
>
> We will follow-up with more discussion of minor questions.
>
> ##  (1)  significance of analysis of required rates for $(\epsilon, \lambda)$
>
> ### 1a) Baseline results for finite-differencing:
> Lines 161-163 of the submission discuss the baseline analysis. We will make this more explicit. CLvdL, Theorem 5 provides a general result on finite differences on the order of, for the case of a uniform kernel, $\left(\epsilon \lambda^{-d_{1}}\right)^{s}+\lambda^{-2}$. To aid discussion we include the statement translated into our notation (which is for multi-point schemes in general, of order $m_0$):
>
> Theorem 5. Suppose regularity conditions hold, that $\delta_{\lambda}(x)$
> is a second-order (product) kernel, and that $\phi_{P}$ has bounded second derivatives in a neighborhood of $x$. Suppose also that, for small enough $\lambda>0$, the parameter $\tilde{\Psi}$ is (m $\left.m_{0}+1\right)$-times pathwise differentiable with gradient of order $m_{0}+1$ evaluated at $P_{\epsilon, \lambda}$ bounded uniformly for sufficiently small $\epsilon>0 .$ Then, with $s:=$ min $(m,$, $m_{0}$ ), we have that
> $\frac{\sum_{j=0}^{m} a_{j, m} \Psi\left(P_{j \epsilon, \lambda}^{*}\right)}{\epsilon}=\phi_{P}(x)+O\left(\epsilon^{s} r_{s+1}(\lambda)^{s+1}+\lambda^{2}\right) .
> $
> (for a uniform kernel, $r_{s+1}(\lambda)=\lambda^{-d_{1} s /(s+1)}$)
>
> This is the baseline rate referenced in line 188, 190-191; 278-280.
>
> The analysis of Thm. 5 in CLvdL applied approximation results for finite differences “off-the-shelf”/generically, i.e. without the exact characterization as we have here or recognizing the role of rate-double-robustness.
>
> Regarding your last question, this previous result does ensure validity for some choice of hyperparameters without requiring a “Thm. 1”-style analysis. In particular, while their improved rates from multi-point schemes also require higher order pathwise differentiability of the estimand, which is understood to be a restrictive condition in the literature on HOIF in causal inference, our improvements by using the statistical structure of the problem do not require higher-order pathwise differentiability but result in similar magnitude improvements in approximation accuracy.
>
> So, we are in your favorable scenario (a). Since these approximation results may not be widely known we will make this clear in the revision by inlining CLvdL thm. 5 (rather than our current reference at 161-163).
>
> ### 1b)
> 1. We will clarify this comparison: the $\epsilon$ dependence arises from the $\epsilon$-dependent nuisance functions, which incur $\epsilon^2$ dependence in Lemma 1.
> 2. The bias in perturbed nuisance is similar to that of Lemma 1. What is different between Prop. 3&4 (vs. Prop 1 or Prop 2) is that the approximation error due to perturbed *matrix inversion* leads to a first-order error dependence (rather than the higher-order $\epsilon$-dependence which arises essentially from repeated application of the product and quotient rules). We will clarify this.

---

> > ### Comment · Reviewer_cYXd · 2022-08-04
> > **Thank you + some clarifications**
> >
> > Thank you, for a very helpful response.  Overall, these clarifications are quite helpful, but I do still have some outstanding clarifications.
> >
> > ## Regarding (1a)
> >
> > Thank you for clarifying that Thm 5 of CLvdL is the "baseline" result: It would be helpful in the paper to not only cite this result, but push all the way through to an "apples-to-apples" comparison.  For instance, as I understand it, $m = 1$ in your context and hence $s = 1$, so that the (numerical) approximation error due to Thm 5 of CLvdL would be $O(\epsilon \lambda^{-d} + \lambda^2)$.  This raises the question: What should this be compared against?
> >
> > Examples of attempting to draw an "apples-to-apples" comparisons: These may be incorrect due to a lack of understanding on my part, but hopefully make the request a bit more concrete.
> > * If I understand correctly, the claim on lines (161-163) of the original version, of the improvement re: dimension dependence, follows from the fact that you consider the squared approximation error in the $O(\epsilon^2 \lambda^{-d})$ term, which would perhaps be something like $O(\epsilon^2 \lambda^{-2d})$ via the analysis of CLvdL?  My guess here may be incorrect, but it would be helpful for clarity if you could inline that general style of comparison.
> > * Perhaps more informative would be to say, following Theorem 1, what requirements on $\epsilon, \lambda$ would be obtained via naive application of Thm 5 of CLvdL, such as comparing $\epsilon \lambda^{-d/2} = o(n^{-max(r\_{\mu},r\_{e})})$ against a different rate that follows from CLvdL.
> >
> > Regarding the last question, you seem to address this in the other comment, so I will save my comment for there.
> >
> > ## Regarding (1b)
> >
> > Your point re: Proposition 2 makes sense, thank you for the clarification.
> >
> > It seems, however, that I should clarify my question on e.g., Propositions 3 & 4.  Here, the reader is meant to compare the two equations, to understand the difference b/w the analytic and empirical derivatives.  The $O(\epsilon)$ additive difference is clear, but there is another difference: The other terms differ in e.g., the use of $V^*(\tilde{s})$ versus $V^*\_\epsilon(\tilde{s})$, and likewise for $d$ versus $d_{\epsilon}$.
> >
> > It would be helpful to at least comment on this: E.g., perhaps it suffices for building intuition to observe the existence of the $O(\epsilon)$ term, without considering the impact of the other differences, but it's not entirely clear to me why.  Perhaps the point is that the difference is *at least* this large, and the generic analysis of CLvdL would similarly yield $O(\epsilon)$?  Or perhaps, to your point, the relevant comparison is not between Propositions 3 & 4, but to compare $O(\epsilon)$ to the results in Propositions 1 & 2?

---

> > > ### Author Response · Authors · 2022-08-04
> > > **reply**
> > >
> > > Thanks for your considerations; we hope these clarifications help further.
> > >
> > > ### Regarding (1a)
> > >
> > > Yes agreed a direct comparison is helpful. We will update the revision shortly but we think the most succinct way to make this comparison is to point out after Thm. 1 that CLvDL require $\epsilon \lambda^{-d / 2}+\lambda^2=o(n^{-1/2})$ (this is essentially what they write in exposition of their Thm. 5).
> > >
> > > ### 1b)
> > >
> > > Lines 662-663 show that $V_\epsilon(s) - V(s)= O(\epsilon)$. By definition of $d(s,a)$ being just a joint distribution, (except for the indicator function) its finite difference approximation is first order in epsilon, $d(s,a)-d_\epsilon(s,a) =\epsilon (d(s,a) - \mathbb{I}(s,a))$. In the nonsmooth case the indicator is a bit tricky, but this (and Lipschitzness of the quotient function for assumed bounded $d(s,a)$) implies that its marginalization is also $O(\epsilon)$.
> > >
> > > Therefore the differences with the nuisance functions doesn't change the qualitative conclusion.
> > >
> > > In the discrete case with smoothing, we also don't incur the dimension dependence as we had in Lemma 1. So, we omitted this discussion to be more explicit in Proposition 3/4. We will also include this explication, or a summary thereof, in explaining the proposition to make the comparison more explicit.

---

> > > > ### Comment · Reviewer_cYXd · 2022-08-05
> > > > **thank you!**
> > > >
> > > > Thank you for the quick response - these clarifications are tremendously helpful, no further questions on this point.

---

> ### Author Response · Authors · 2022-08-02
> **response to reviewer cYXd, 2)**
>
>
> ## 2) Unclear significance of limitations of Empirical Gateaux derivatives:
>
> Thank you for pointing out that our discussion of limitations is also an important opportunity to situate this framework in the broader literature, to understand how others’ results mitigate some of these limitations. We propose to move our discussion of limitations, including the following invocations of sufficient conditions from the literature,  from the conclusion (as a Neurips-style statement of limitations) to the end of section 2, which will help introduce these more general conditions.
>
> ### 1. Are there existing scenarios where the analytic form of the gateaux derivative is non-obvious, but where … pathwise differentiability, second-order remainder … can nonetheless be verified to hold?
>
> There are high-level primitive conditions in the research literature for both pathwise differentiability/asymptotic linearity; or second-order remainder terms. We will restate these in the main text for clarity. (We omitted discussion of these results because we do not further contribute to them, but agree that it would be helpful for exposition).
>
>  To establish that the remainder term is second-order in general, Kirtheswamy et al. assume stronger Frechet differentiability in L2 norm (discussion above eq. 4, pg 3) and correspondingly show that the remainder term is second-order in the error of the densities.
>
> For asymptotic linearity (pathwise differentiability), Ichimura and Newey 2015, section 4 provide primitive sufficient conditions for asymptotic linearity of nonparametric functionals, again under stronger differentiability assumptions, that the Frechet derivative is Lipschitz, showing that the remainder term will be second order again under rate assumptions on the underlying densities (e.g. assumption 2). Although Frechet differentiability is indeed stronger than Gateaux differentiability, Kallus/Mao/Uehara (Localized Debiased Machine Learning) show that it is satisfied by estimands considered in causal inference such as the ATE).
>
> These general sufficient conditions also apply to settings where the influence function is non analytically available.
>
>
>
> ### are there scenarios where we can apply this approach (with appropriately conservative rates on (ϵ,λ)?
> Yes, thm. 5 of CLvdL provides rates on $(\epsilon,\lambda)$ where the approach can be applied in general. Our work shows how specializing to the statistical structure of the problem can weaken these rate requirements from generic, conservative ones. Yes,  the constrained MDP with arbitrary linear constraints is an example.
>
>
> ### Comments on soundness:
> - Lemma 1: this is an application of the triangle inequality with respect to $\tilde\mu_{\epsilon} (X) - \tilde{\mu}(X) + \tilde{\mu}(X) - \mu(X)$. (Although triangle inequality can be proved by Cauchy-Schwarz we will edit this to be clearer that we are using a triangle inequality with L_{2,P}).
> - line 562:
> Thanks for pointing out that we combine a number of steps here. For clarity we will separate these out.
> The steps in the inequality are: lower bound assumption on the covariate density, upper bound probability density by 1, and identify the kernel variance term as $\lambda^{-d} \int \tilde{\delta}_{\lambda}^2(u) du$; the last term is a kernel-dependent term that is some constant.
> Unfortunately, different sources use different definitions for kernel roughness. To avoid any confusion we will refer to this as the kernel variance term. (The $\lambda^{-d}$ term arises from a change of variables applied to $\tilde{\delta}_\lambda$; dimension dependence from assumption that is a product kernel over dimensions, and general assumption of constant kernel-dependent normalized integral). See, e.g. Pagan and Ullah 24-26.

---

> > ### Comment · Reviewer_cYXd · 2022-08-04
> > **Further clarifications**
> >
> > ## Regarding (2) / "scenarios where we can apply this approach with conservative rates"
> >
> > Perhaps my question was not entirely clear (or perhaps I misunderstand the response): I meant to ask "are there problems where we can *prove* that these conditions (pathwise differentiability / bounded second-order remainder) hold, without analytically deriving the influence function", while you seem to have provided examples of *additional assumption* that could be made (e.g., Frechet differentiability), which would then imply that these conditions hold.
> >
> > E.g., I was hoping for an answer like "One can prove, in Example 3, that these conditions hold for any set of constraints $\mu \in \mathcal{P}$, without additional assumptions", even though, for a particular set of constraints, the influence function would change.
> >
> > ## Regarding the triangle inequality in Lemma 1
> >
> > Apologies for belaboring the point, but this does not address my concern. From what I understand, your argument is that $E[(X + Y)^2] \leq E[X^2] + E[Y^2]$, where $X = \tilde{\mu}_{\epsilon} - \tilde{\mu}$ and $Y = \tilde{\mu} - \mu$. This is not true in general: For example, if $X = 1$ and $Y = 1$ almost surely, this would yield the conclusion that $4 \leq 2$.
> >
> > The lack of square roots appears to be the missing piece here. As I understand it, the relevant triangle inequality argument would be $\sqrt{E[(X + Y)^2]} \leq \sqrt{E[X^2]} + \sqrt{E[Y^2]}$. The full logic would be:
> > * $L_{2, P}$ is equipped with an inner product given by $\langle X, Y \rangle = E[XY]$
> > * The corresponding norm is given by $||X|| = \sqrt{\langle X, X \rangle} = \sqrt{E[X^2]}$, which also defines a metric.
> > * The resulting triangle inequality is that $||X + Y|| \leq ||X|| + ||Y||$, or equivalently that $\sqrt{E[(X + Y)^2]} \leq \sqrt{E[X^2]} + \sqrt{E[Y^2]}$.
> >
> > Perhaps I am missing something here, or perhaps there is indeed an error, but one that doesn't meaningfully alter the results?  A clarification either way seems necessary.

---

> > > ### Author Response · Authors · 2022-08-04
> > > **reply**
> > >
> > > Thanks for comment and your careful analysis, we hope this clarifies further.
> > >
> > > ## Regarding (2) / "scenarios where we can apply this approach with conservative rates"
> > >
> > > We would also hope that this may be possible. What we suggest in this response is to elucidate and collect results that showcase stronger conditions (Frechet, not Gateaux) that can admit pathwise differentiability, and point to discussions that instantiate these stronger conditions. In doing so, we anticipate further work can widen the class of problems, building on this characterization. We're not trying to claim more beyond that at this point.
> > >
> > > To sum the overall message we intend with these addendums, "yes for the case of quadratically regularized convex optimization, asymptotic linearity in general has been established. More generally, not exactly, but the tools are there for this to improve in the future. There is a path that would be important future analysis.". We agree with you overall that this is a crucial direction for future study.
> > >
> > > A general class of problems ```... where we can prove that these conditions (pathwise differentiability / bounded second-order remainder) hold, without analytically deriving the influence function''' are those of strongly convex stochastic programs: Proposition 1 of Ruan and Duchi's Asymptotic Optimality in Stochastic Optimization rewrites results in Bonnans and Shapiro's "Perturbation Analysis of Optimization Problems" and establishes asymptotic linearity under primitive conditions of strong convexity. Again, analogous to moving the assumption of Frechet differentiability, this establishes pathwise differentiability under stronger structural assumptions. But, it is easier to reason about strong convexity than Frechet/Gateaux differentiability. A reasonably knowledgeable convex optimizer can tell if something is strongly convex or not, whereas appealing to Frechet differentiability rather than Gateaux also essentially requires seasoned researchers (who can also derive influence functions, so not the intended users of such approaches) to verify these conditions for larger classes of problems. We include these conditions not to suggest that analysts can do the latter, but to encourage researchers to do so.
> > > We didn't include this in the main text because it's a tangent, and our optimization analysis is of linear programs without quadratic regularization, but agree that adding these elucidations in the appendix can help.
> > >
> > >
> > > Our appeal to asymptotic linearity under Frechet differentiability indeed move the assumption elsewhere, onto the (stronger) smoothness of the functional. (This is fairly common in causal inference, to move untestable assumptions to other (untestable) aspects). These two discussions together -- strengthening the smoothness assumptions of the functional, and pointers to previous literature establishing that this in fact holds -- therefore point to classes of problems where this can be applied. We referenced Kallus, Mao, Uehara's LDML who had also assumed Frechet differentiability. Their section 1.3 on "Solving an estimating equation under incomplete data" discusses a general form of estimating equation with incomplete data that satisfies Frechet differentiability.) We omitted these elaborations from the main text due to space concerns but can add them to the appendix. To the best of our knowledge, drawing this conclusion from these two discussions hasn't been written down, and will provide useful pointers to build on.
> > >
> > > Pathwise differentiability is essentially a fundamental limitation of attempts to "automate" semiparametrics because it is a condition that requires an amount of training to understand that is on the order of deriving an influence function anyway. (This is a major point in Luedtke et al.'s "Discussion of "Deductive derivation and turing-computerization of semiparametric efficient estimation" by Frangakis et al"). While primitive conditions of strong convexity are possible for analysts to check, we agree that Frechet differentiability isn't quite; the hope there is instead that researchers will also establish Frechet differentiability for a larger class of functionals. Currently, to the best of our knowledge, all works in the small area of automatic semiparametrics make this assumption. Most of causal inference is moving around untestable assumptions to other aspects which may be easier to reason about or build upon: our elucidations simply provide a few different concrete options.
> > >
> > >
> > > ## the triangle inequality in Lemma 1
> > >
> > > We had meant triangle inequality with RMSE. It doesn't meaningfully alter the results at all to apply the same analysis with RMSE (which is a norm, satisfying the triangle inequality). The product rate conditions simply hold with respect to RMSE rather than MSE (i.e. product rate of unperturbed nuisances is $n^{-\frac{1}{2}}$ consistent, and the bounds on the perturbed term go through with square roots). Thanks for noting.

---

> > > > ### Comment · Reviewer_cYXd · 2022-08-05
> > > > **thank you!**
> > > >
> > > > Thank you again for the quick response.  The broader discussion provided re: "general classes of problems" is compelling, and agree that it could serve as an appendix section.  I can also see the argument re: RMSE vs. MSE for the triangle inequality, and agree that it wouldn't impact the results.
> > > >
> > > > All my concerns have been addressed by the discussion, and I'm convinced that the relevant clarifications would be straightforward to make.  With that in mind, I will increase my score.

---

### Official Review · Reviewer_egsh · 2022-07-13

**Rating:** 4
**Confidence:** 1
**Soundness:** 2 fair
**Presentation:** 2 fair
**Contribution:** 2 fair

**Summary:**

This paper discusses the Gateaux derivative for a parameter functional of a probability distribution. The authors propose a method for approximating the Gateaux derivative empirically, with applications to causal inference. The authors also suggest dynamic treatment regimes as an interesting application.

**Questions:**

Following on from the above question, I have the following minor question.
- Is it possible to apply the proposed method to robust statistics, such as Koh and Liang (2017)?
- I think one of the parentheses in Lemma 1 is not closed.
- What estimator (method) is used to estimate the nuisance parameter in the counterfactual mean estimation in Section 4.1?

**Limitations:**

See the above comments.

**Strengths And Weaknesses:**

First, I could not fully understand this study. In particular, I would like the authors to elaborate on the motivation more. Therefore, I would like to make a final evaluation after a discussion with the authors.

I think that the authors' proposed method for approximating the Gateaux derivative is interesting. As the authors mention, the Gateaux derivative and the influence function play an important role in statistical inference. For example, in (semiparametric) counterfactual mean estimation, we usually analytically derive the (semiparametric) influence function and use it as a lower bound of the asymptotic variance; then, we propose an estimator that achieves the lower bound. However, I could not understand what it means to approximate the Gateaux derivative empirically, and I would appreciate a careful explanation.

The influence function is helpful in constructing the estimator, but I do not think it is necessary to be concerned with its estimation as long as it can be analytically derived. For example, Hirano, Imbens, and Ridder (2003) propose an IPW-based estimator that achieves the lower bound derived from Hahn (1998)'s (AIPW-style) influence function. Of course, defining the influence function for a broad class of estimators is an important issue, and several previous studies have attempted it. However, once the Gateaux derivative is defined, it is not clear to what extent its empirical approximation needs to be discussed. At least, it does not seem to be necessary for counterfactual mean estimation and many OPE tasks.

In summary, while I acknowledge the contribution of the method, I would like to know more about the motivation.

---

> ### Author Response · Authors · 2022-08-02
> **Response to Reviewer egsh: elaboration on motivation**
>
> ### Clarifying motivation (R1)
>
> We will add this following paragraph walking through the motivation and significance in the introduction more concretely, before the contributions paragraph:
>
> ```
> Our analysis and concretizations illustrate the exact relationship between the numerical approximation and the analytical derivative. The numerical approach is particularly useful in settings where pathwise differentiability is established for a class of estimators, such as constrained optimization. Previous influence function adjustments work instead with closed-form solutions of special cases of optimization problems: we later show how the finite-difference argument naturally generalizes to analyst-specified custom constraints. However, in general, it is well-known that although finite-difference approaches require small discretizations, finer-scaled discretization run into fundamental issues with floating-point computation. Therefore, our analysis, which shows how the statistical structure of the target estimand weakens approximation error requirements for accuracy, is useful in expanding the computational regimes where numerical differentiation is applicable.
> ```
>
> ### "However, I could not understand what it means to approximate the Gateaux derivative empirically, and I would appreciate a careful explanation."
>
> We clarify the explanation we give in lines 84-85. The analytical Gateaux derivative computes the derivative analytically (i.e. of the perturbed densities, and simplifies the integral). The numerical Gateaux derivative is, analogously, the finite-difference numerical approximation of the single-variable derivative of the perturbed densities. (The pathwise derivative is defined as the single-variable derivative of this perturbation). However, previous approaches for computing the numerical derivative have implicitly assumed all the probability distributions were known: integrals were computed with known probability distributions. Hence, for use for statistical approximation, these densities used to compute the integral must also be estimated, e.g. rather than assuming $P(Y,A=1,X)$ is known, it is estimated. We call this last approach the “empirical Gateaux derivative”.
>
> Lines 82-84 discuss this. We will expand the explanation further, e.g. revising our description to
>
> ```...empirical Gateaux derivative computed by numerical approximation with estimated probability densities $\tilde P(Y,A=1,X)$, the numerical Gateaux derivative obtained by finite-differencing but with the true probability densities $P(Y,A=1,X)$ which are generally unavailable in the estimation setting, ...```
>
> The code (and analysis) for CLvdL assumed the density estimates were known. An approach for automatic improvement of statistical estimators cannot actually assume that the models (densities or conditional expectations) are fully known. Therefore we study when these components of the model need to be estimated from data.
>
>  ### "However, once the Gateaux derivative is defined, it is not clear to what extent its empirical approximation needs to be discussed. At least, it does not seem to be necessary for counterfactual mean estimation and many OPE tasks."
>
> We will clarify that our characterizations are intended to illustrate precisely what the general method is doing, in a setting where we have a well-understood target estimator, and hence can exactly compare the resulting empirical and numerical approximations. This theoretical characterization is important so that methodologists / researchers / analysts can better understand these techniques.
> We build on these initial case studies with well-understood analytical Gateaux derivatives, where the analytical Gateaux derivatives are well-known, to our final case study with constrained Markov Decision Processes, to show how this approach seamlessly transitions to this class of “custom” estimators (see our generalization to constraints, lines 281-290), without restricting to constraints that lead to closed-form solutions, as was the case previously with analytical Gateaux derivatives.
>
> We hope this elaboration of the motivation and elaboration of the discussion included in the paper, and that you could please evaluate the paper based on its contributions.

---

> ### Author Response · Authors · 2022-08-03
> **response to minor questions**
>
> ### Is it possible to apply the proposed method to robust statistics, such as Koh and Liang (2017)?
>
> As mentioned in lines 323-324, the main difference is that we target estimation of a statistical functional, whereas Koh and Liang seek to compute the influence function of the machine learning model itself. Therefore, the targets of estimation are very different. We agree it would be interesting to build connections.
>
> ### What estimator (method) is used to estimate the nuisance parameter in the counterfactual mean estimation in Section 4.1?
>
> We use kernel density estimates (equiv. Nadaraya-Watson regression). We will add a note to clarify.

---

> ### Author Response · Authors · 2022-08-08
> **(polite ping)**
>
> Thanks again for your review. Please do let us know if you have any further questions that we could clarify while we are able to do so.

---

### Author Response · Authors · 2022-08-03
**rebuttal revision**

Thanks again for the comments. We have incorporated comments and uploaded a revision. Changes are marked in *blue* (minor changes are not demarcated in blue). (For the most part, these summarize responses to reviewers' main concerns, that we had included in individual responses to reviewers. Additional incorporation of responses will be included in the appendix.)

---

### Meta-Review · Area_Chair_NiN3 · 2022-08-27

**Recommendation:** Accept
**Confidence:** Certain

**Metareview:**

The authors make a solid contribution in the literature on computerized estimation of gateaux derivatives and automatic debiasing, with applications to causal inference, providing theorems on the level of numerical approximation that preserves root-n statistical rates. Therefore this should be an interesting paper for the causal ml community. The authors have addressed most major concerns raised by reviewers in their original evaluation.

On a minor note the authors should also relate to the recent prior work on automatic debiased machine learning for dynamic effects https://arxiv.org/abs/2203.13887 which seems to be capturing their main application, contrary to what is claimed in their related work.

**Award:**

No

---

### Decision · Program_Chairs · 2022-09-14

Accept